# Modulation of Pulmonary Fibrosis by Pulmonary Surfactant-Associated Phosphatidylethanolamine In Vitro and In Vivo

**DOI:** 10.3390/ijms26157132

**Published:** 2025-07-24

**Authors:** Beatriz Tlatelpa-Romero, Luis G. Vázquez-de-Lara Cisneros, Olga Cañadas, Amaya Blanco-Rivero, Barbara Olmeda, Jesús Pérez-Gil, Criselda Mendoza-Milla, José Luis Martinez-Vaquero, Yair Romero, David Atahualpa Contreras-Cruz, René de-la-Rosa Paredes, Sinuhé Ruiz-Salgado, Roberto Berra-Romani, Alonso Antonio Collantes-Gutiérrez, María Susana Pérez-Fernández, María Guadalupe Hernández-Linares, Gabriel Guerrero-Luna

**Affiliations:** 1Laboratorio Central de Investigación, Facultad de Medicina, Benemérita Universidad Autónoma de Puebla, Puebla 72420, Mexico; beatlarom@outlook.com (B.T.-R.); joseluis.martinez@upaep.edu.mx (J.L.M.-V.); 2Laboratorio de Medicina Experimental, Facultad de Medicina, Benemérita Universidad Autónoma de Puebla, Puebla 72420, Mexico; 3Departamento de Bioquímica y Biología Molecular, Facultad de Ciencias Biológicas, Universidad Complutense de Madrid, 28040 Madrid, Spain; ocanadas@quim.ucm.es (O.C.); amayabla@ucm.es (A.B.-R.); barbara_olmeda@bio.ucm.es (B.O.); jperezgil@bio.ucm.es (J.P.-G.); 4Instituto de Investigación Hospital 12 de Octubre (imas12), 28041 Madrid, Spain; 5Laboratorio de Transducción de Señales, Unidad de Investigación, Instituto Nacional de Enfermedades Respiratorias Ismael Cosío Villegas, Ciudad de México 14080, Mexico; criselda.mendoza@iner.gob.mx; 6Laboratorio de Biopatología Pulmonar de Enfermedades Fibrosantes, Facultad de Ciencias, Universidad Nacional Autónoma de México, Ciudad de México 04510, Mexico; yair12@hotmail.com; 7Unidad Multidisciplinaria de Investigación Experimental, Laboratorio de Síntesis Farmacéutica, Facultad de Estudios Superiores Zaragoza, Universidad Nacional Autónoma de México, Batalla 5 de Mayo s/n Esquina Fuerte de Loreto, Ejército de Oriente, Iztapalapa 09230, Mexico; 8Hospital General de Puebla “Dr. Eduardo Vázquez Navarro”, Puebla 72490, Mexico; rdelarosa2000@hotmail.com; 9Área Académica de Ciencias de la Tierra y Materiales, Universidad Autónoma del Estado de Hidalgo, Pachuca 42184, Mexico; snhruiz@gmail.com; 10Laboratorio de Fisiología Cardiovascular, Facultad de Medicina, Benemérita Universidad Autónoma de Puebla, Puebla 72420, Mexico; roberto.berra@correo.buap.mx; 11Hospital Universitario de Puebla, Puebla 72410, Mexico; alonso.collantesg@correo.buap.mx (A.A.C.-G.); susana.perez@correo.buap.mx (M.S.P.-F.); 12Laboratorio de Flujo Continuo y Fotoquímica, Centro de Química, Instituto de Ciencias BUAP, Puebla 72570, Mexico

**Keywords:** native porcine pulmonary surfactant, phosphatidylethanolamine, surface tension, biophysical evaluation, lung lavages, pulmonary fibrosis

## Abstract

Pulmonary fibrosis (PF) is characterized by excessive collagen deposition and impaired lung function. Pulmonary surfactant may modulate fibroblast activity and offer therapeutic benefits. We developed a natural porcine pulmonary surfactant (NPPS) enriched with 1,2-dipalmitoyl-rac-glycero-3-phosphatidylethanolamine (PE) and evaluated its biophysical and biological properties. Biophysical analysis showed that PE improved surfactant performance by increasing surface pressure and stability. In vitro, NPPS-PE reduced collagen expression and induced apoptosis in normal human lung fibroblasts; in addition, it decreased proliferation in fibroblasts stimulated with TGF-β. In vivo, NPPS-PE improved gas exchange and significantly reduced collagen deposition in bleomycin-treated mice. These findings suggest that NPPS-PE may be a promising therapeutic strategy for fibrosing lung diseases.

## 1. Introduction

The lung is a dynamic organ that continuously interacts with the environmental air introduced through processes of compression and expansion [1]. This activity, combined with constant exposure to environmental pollutants, predisposes it to diseases such as pulmonary fibrosis (PF) [2]. PF is a progressive disease characterized by excessive deposition of a collagen-rich extracellular matrix by fibroblasts, leading to irreversible scarring and impaired oxygen exchange in the alveoli [3,4]. PF often results from a diverse group of conditions collectively known as diffuse parenchymal lung diseases or interstitial lung diseases, which can progress to fibrosis in chronic cases. These conditions include silicosis, asbestosis, and drug- and radiation-induced lung diseases. Others include granulomatous disease such as sarcoidosis and idiopathic pulmonary fibrosis (IPF), the cause of which remains unknown [5,6].

Given the role of pulmonary surfactant in maintaining alveolar stability, alterations in its composition may have profound implications for diseases such as PF. Pulmonary surfactant, a complex mixture synthesized, stored, and recycled by type II alveolar cells, plays a critical role in maintaining lung function [7,8]. Pulmonary surfactant is composed of 10% specific proteins, such as SP-A, SP-B, SP-C, and SP-D, and 90% lipids, which are primarily phosphoglycerides, cholesterol, and neutral lipids [9,10]. The main function of Pulmonary surfactant is to reduce the surface tension in the alveoli, which minimizes the work of breathing and prevents alveolar collapse during expiration [11]. Alterations in the composition or deficiency of pulmonary surfactant have been associated with major lung diseases, including neonatal respiratory distress syndrome, chronic obstructive pulmonary disease, and IPF [12,13].

Recent studies suggest that pulmonary surfactant components, particularly 1,2-dipalmitoyl-rac-glycero-3-phosphatidylethanolamine (PE), may modulate fibroblast behavior and halt the fibrotic process before PF develops. For instance, Beractant, a pulmonary surfactant formulation, was shown to induce apoptosis in normal human lung fibroblasts (NHLF), decrease collagen expression, and trigger calcium signaling, an effect associated with antifibrotic properties [14]. In contrast, SP-A increased collagen expression without affecting collagenase-1 or tissue inhibitor of metalloproteinases-1. Interestingly, when Beractant was combined with SP-A, the antifibrotic effects were partially reversed. These findings suggest that pulmonary surfactant lipids may protect against fibrogenesis by promoting fibroblast apoptosis and reducing collagen accumulation [15]. In addition, PE has been shown to have significant antifibrotic effects. In a bleomycin-induced mouse model of lung fibrosis, PE reduced collagen expression, promoted apoptosis, and attenuated overall fibrosis [16]. Similarly, organically synthesized PE induced early apoptosis at low concentrations, significantly increased late apoptosis at higher doses in NHLF, further strengthening its potential as a therapeutic target for PF [17]. Considering the above, the aim of this study is to present the biophysical properties and the antifibrotic effects of a porcine-derived pulmonary surfactant enriched with PE obtained through organic synthesis [17]. Evidence of its surfactant properties and its effects on pulmonary fibrosis are presented, both in vitro and in vivo.

## 2. Results

### 2.1. Surface Activity of NPPS-PE Mixtures

To analyze the organization of the different lipid phases also in interfacial films, NPPS monolayers were subjected to five cycles of compression/expansion, resulting in a shift of the compression isotherms to lower area values (Figure 1A–F), indicating material loss. The material expelled from the interfacial film could not be re-incorporated into the monolayer during expansion. This led to significant hysteresis and a progressive decrease in the maximum and minimum surface pressures with each cycle (Figure 1G and H, respectively). The decreasing maximum surface pressure across cycles suggests that compression causes a loss of dipalmitoylphosphatidylcholine molecules. In addition, the NPPS isotherms displayed a plateau at a surface pressure of approximately 42 mN/m, which is indicative of the refinement of the interfacial film composition due to reservoir formation.

The addition of increasing amounts of PE produced stable monolayers with greater hysteresis than that observed in NPPS films alone (Figure 1A). The isotherms showed multiple plateaus: one near 40 mN/m, likely associated with reservoir formation, and another at surface pressures exceeding 60 mN/m, suggesting the collapse of the DPPC-enriched monolayer. The incorporation of increasing amounts of PE into the NPPSmonolayer increased the length of this second plateau (Figure 1B–E). It also increased the maximum pressure values reached in each cycle (Figure 1G), suggesting that PE promotes the retention of PC or exhibits further surfactant activity due to its saturated acyl chains. To evaluate the potential intrinsic surface activity of PE, compression/expansion cycles were performed on pure PE monolayers. The results (Figure 1F) demonstrated that PE forms stable monolayers capable of also reducing the surface tension to low values with minimal film compression (Figure 1H).

### 2.2. Molecular Organization of NPPS and PE in Interfacial Films

We used the Langmuir–Blodgett technique to form interfacial films, which were transferred onto solid supports to analyze compression-driven segregation of two-dimensional domains by epifluorescence microscopy. At low surface pressures, pure NPPS monolayers were uniformly stained red because of the presence of the fluorescent lipid rhodamine-PE partitioned homogeneously into the LE phase (Figure 2, top row). As the pressure increased above 4 mN/m, small dark areas appeared, representing the nucleation and segregation of condensed domains and LE/LC phase coexistence. Increasing the pressure further, both the number and size of these domains apparently increased. In contrast, pure PE monolayers remained in the LE phase up to 13 mN/m, where very small LC domains began to appear (Figure 2, lower row). When pressure was raised above 30 mN/m, the size of the LC domains slightly increased, without affecting their number. Upon incorporation of PE into NPPS monolayers, further fluidization occurred, and higher surface pressures (13 mN/m) were required to observe LC domains that were larger than those seen without PE (Figure 2).

### 2.3. Surface Tension and Surfactant Behavior of NPPS-PE Under Compression-Expansion Dynamics

The effect of PE on the ability of NPPS to reduce surface tension (ү) to very low values was evaluated at different PE molar ratios (1/0, 1/0.1, 1/0.3, 1/0.5, 1/1, and 0/1) using a captive bubble surfactometer. Surface tension monitoring was performed under several conditions, including initial and post-expansion adsorption, quasi-static cycling, and dynamic compression–expansion cycling (comparing the first, tenth, and twentieth cycles), both in the presence and absence of PE. The results of these experiments are presented as surface tension versus time isotherms, right after application (initial adsorption) and after a rapid expansion (post-swelling adsorption), and as surface tension versus area fraction isotherms upon compression–expansion cycling. Data are averaged from three experiments, with standard deviation plotted for clarity.

NPPS significantly reduced the surface tension in less than one second under both static (initial adsorption) and post-expansion conditions, indicating rapid interfacial adsorption (Figure 3, first row left and center-left panels, respectively). It took approximately 20 s to reach an equilibrium tension of 25 mN/m. The addition of increasing amounts of PE did not affect the ability of NPPS to rapidly adsorb into the air/water interface under any of the conditions studied (Figure 3, rows 2–5 left and center-left panels, respectively). Two compression/expansion cycling regimes were used to assess the effect of PE on the surface behavior of NPPS: a quasi-static regime, which provides insight into the lipid exclusion process occurring during compression, and a dynamic regime, which provides information on the surface activity under conditions that mimic respiratory cycles. During the quasi-static cycles, a refinement in the composition of NPPS occurred during the first cycle, which is necessary for NPPS to reach surface tension values around 2 mN/m with minimal compression in the subsequent cycles (Figure 3, first row, center-right panel). This refinement in the composition was not observed in the dynamic cycles since no hysteresis occurred in any of the 20 cycles performed (Figure 3, first row, right panel). The incorporation of increasing amounts of PE produced a slight increase in hysteresis during the quasi-static cycles but did not affect the compression required to reduce surface tension to values close to 0 mN/m (Figure 3, rows 2–5 center-right panels) nor the behavior of NPPS during dynamic cycling (Figure 3, rows 2–5, right panels).

### 2.4. Surfactant Deficiency in Rats and Guinea Pigs

The PS deficiency model was developed by connecting the trachea of the animal to a positive pressure ventilator, which was attached to an oxygen tank under constant monitoring (Figure 4A). No criteria for including or excluding animals were used. In the control group (n = 3), the baseline oxygen saturation (SpO_2_) at time point 0 was 97.5 ± 1.5%. Following lung lavages as indicated in the Material and Methods section, the SpO_2_ at subsequent time points was 35 ± 0%, 35 ± 0%, 42.94 ± 7.94%, 38.93 ± 3.93%, 37.46 ± 1.46%, 41.31 ± 6.31%, and 53.50 ± 11.50% at time points of 30, 40, 50, 60, 70, 80, and 90 min, respectively. In the NPPS-PE treatment group (n = 3), the SpO_2_ at time point 0 was 96.33 ± 3.06%. Following lung lavages, the SpO_2_ at time points 30, 40, 50, 60, 70, 80, and 90 min was 39.0 ± 8.19%, 91.67 ± 6.66%, 77 ± 19.47%, 83.33 ± 18.45%, 81 ± 18%, 73.33 ± 7.37%, and 74 ± 4.58%, respectively (Figure 4A).

### 2.5. NPPS-PE Prevents Cell Proliferation, Induces Apoptosis, and Reduces Collagen I Expression in NHLF

NPPS-PE significantly inhibited cell proliferation in NHLF, as determined by WST-1 assay, indicating a potential antiproliferative effect: TGF-β was 67.72 ± 7.12%, NPPS 49.47 ± 4.35% (*p* < 0.001 **), PE 53.43 ± 2.33% (*p* < 0.001 **), and NPPS-PE 48.15 ± 7.33% (*p* < 0.0001 ***). At 48 h, the growth rate for TGF-β was 96.65 ± 3.28%, while NPPS was 49.07 ± 4.35% (*p* < 0.001 **), PE 50.03 ± 3.24% (*p* < 0.001 **), and NPPS-PE 35.04 ± 5.07% (*p* < 0.0001 ***) (Figure 5).

In IPF, fibroblasts exhibit resistance to apoptosis, a critical process for proper epithelial repair. Apoptosis can be assessed at different stages of cell death. One of these stages involves the exposure of phosphatidylserine at the outer leaflet of the plasma membrane, which can be detected using annexin V. In addition, propidium iodide (PI) binds to the genetic material of cells with compromised membranes, making it useful for identifying late-stage apoptosis. Both fluorochromes are commonly used in flow cytometry to detect apoptotic cells.

For this study, 50,000 cells were seeded in 24-well plates and exposed to NPPS, PE, or NPPS-PE treatments for 24 and 48 h. Apoptosis was evaluated by flow cytometry, allowing the quantification of early apoptosis in the Q2-UR panel and late apoptosis in the Q2-LR panel (Figure 6A). The sum of the early and late apoptosis quadrants was added to give us the following values. In the control group, apoptotic cells accounted for 1.94 ± 1.79% at 24 h and 0.40 ± 0.40% at 48 h. In contrast, the NPPS-treated group exhibited significantly higher apoptosis rates, reaching 28.67 ± 3.49% at 24 h and 54.83 ± 5.30% at 48 h (*p* < 0.0001). Similarly, the apoptosis in the PE-treated group was 37.14 ± 4.41% at 24 h and 55.19 ± 2.46% at 48 h (*p* < 0.0001). The NPPS-PE-treated group showed the highest apoptosis rates, with 41.41 ± 3.56% at 24 h and 61.12 ± 8.64% at 48 h (*p* < 0.0001) (Figure 6B).

Early apoptosis is detected when annexin V binds to phosphatidylserine that has translocated to the outer membrane of the cells, resulting in green fluorescence. In contrast, late apoptosis is detected through the binding of propidium iodide to exposed genetic material following membrane degradation (Figure 6C).

### 2.6. Reduced Col1A1 Expression in NHLF Treated with NPPS-PE

This study evaluated Col1A1 collagen gene expression in three NHLF cell lines under different experimental conditions: exposed to NPPS, PE, or NPPS-PE at 24 and 48 h. The expression levels of Col1A1 were assessed using RT-qPCR and the 2^−ΔΔCT^ method, which calculates relative gene expression based on the Ct values of the samples normalized to those of the control. The endogenous gene GAPDH served as an internal control.

At 24 h, the average expression of Col1A1 was as follows: control group: 9 ± 0.49 2^−ΔΔCT^; NPPS group: 10.61 ± 0.36 2^−ΔΔCT^; PE group: 10.47 ± 1.02 2^−ΔΔCT^; and NPPS-PE group: 7.72 ± 0.93 2^−ΔΔCT^. Statistical analysis revealed significant differences between the NPPS and NPPS-PE groups (*p* < 0.0001 ***), as well as between the control and NPPS-PE groups (*p* < 0.001 **). At 48 h, the average expression of Col1A1 in the control group was 9 ± 0.71 2^−ΔΔCT^, in the NPPS group, it was 9.16 ± 0.52 2^−ΔΔCT^, in the PE group, it was 3.41 ± 0.14 2^−ΔΔCT^, and in the NPPS-PE group, it was 3.15 ± 0.67 2^−ΔΔCT^. Statistical analysis showed significant differences between the NPPS and NPPS-PE groups (*p* < 0.0001 ***), between the NPPS and PE groups (*p* < 0.01 *), and between the control and NPPS-PE groups (*p* < 0.001 **) (Figure 7).

### 2.7. NPPS-PE Attenuates the Effect of Bleomycin During the Inflammatory Stage in Mice

To evaluate the potential protective role of NPPS-PE during the inflammatory phase of bleomycin-induced lung injury in mice, hydroxyproline content and collagen deposition were quantified as indicators of fibrotic response.

Hydroxyproline in the control group (n = 9) was 2.31 ± 0.55 µg/mg compared to bleomycin (n = 6), which was 11.68 ± 3.39 µg/mg, *p* < 0.001 ***. NPPS-PE on day 0 (n = 6) was 2.52 ± 0.84 µg/mg, NPPS-PE on day 3 (n = 6) was 7.77 ± 0.92 µg/mg, *p* < 0.001 ***, and NPPS-PE on day 7 (n = 6) was 8.80 ± 3.19 µg/mg. The percentage of collagen in the tissue for the control group was 13.57 ± 4.13%, while for the bleomycin group, it was 29.99 ± 4.23%, *p* < 0.0001 ***. NPPS-PE on day 0 showed 20.23 ± 3.08%, *p* < 0.001 **, NPPS-PE on day 3 showed 20.70 ± 4.52%, *p* < 0.001 **, and NPPS-PE on day 7 exhibited 25.41 ± 3.07% (Figure 8).

## 3. Discussion

In this study, an NPPS formulation was obtained with a phospholipid and protein composition similar to other pulmonary surfactants commercially available. Once enriched with PE (NPPS-PE), the biophysical evaluation showed a synergistic activity of PE on NPPS surface activity. Furthermore, the effect of the NPPS-PE mixture in NHLF was found to induce an antifibrotic action, as previously observed in experiments with commercial substances [16]. We also provide evidence that NPPS-PE improves oxygenation in an animal model of surfactant deficiency.

The biophysical function of pulmonary surfactant is to reduce surface tension in the lungs, helping prevent alveolar collapse [8]. A good surfactant must fulfill three essential functions for respiration: forming an active film during inhalation, reorganizing to reduce surface tension during exhalation, and redistributing lipids in the next inhalation [18,19]. The transition from a liquid-expanded phase to a tilted-condensed phase is critical, as the tilted-condensed phase favors the reduction of surface tension [19,20]. The addition of PE improved surfactant activity. Using the Langmuir film balance, NPPS alone showed a typical surfactant compression–expansion isotherm [21,22]. When PE was added, it showed an optimal maximum surface tension of ~70 mN/m and collapsed near ~10 mN/m, comparable to previous studies [23]. Domain segregation was observed, indicating potential for compression-driven lipid-depuration and reintegration into type II epithelial cells [8,24,25]. PE facilitated domain segregation in a concentration-dependent manner, likely due to a synergistic action. Captive bubble surfactometer analysis confirmed that PE optimizes adsorption dynamics.

The in vivo findings in surfactant deficiency models reflected those observed in vitro. In rats and guinea pigs, NPPS-PE administration increased oxygenation and O_2_ saturation, consistent with previous studies [26,27]. The biochemical composition of NPPS was similar to commercial surfactants. TLC revealed the presence of key phospholipids and cholesterol (Appendix A), including PE at ~7%, consistent with other porcine-derived surfactants [28].

In NHLF, NPPS-PE reduced collagen expression and increased apoptosis. Pulmonary fibrosis results from excessive fibroblast proliferation and extracellular matrix accumulation [2,29]. Fibroblasts in fibrotic tissue resist apoptosis, perpetuating tissue remodeling [15,17,30,31]. In a bleomycin-induced pulmonary fibrosis model, NPPS-PE reduced collagen deposition and fibrosis, especially during the inflammatory phase. Inhalation delivery targeted the lungs directly. Optimal treatment timing was explored, as inflammation begins immediately post-bleomycin [32]. This phase involves cytokine recruitment, immune cell infiltration, and epithelial damage, promoting fibroblast activation via macrophage signaling [4,33]. NPPS-PE may integrate into alveolar coating films, modulating inflammation and restoring biophysical function. Surfactant dysfunction may precede fibrosis and serve as an early marker [34]. Timely NPPS-PE treatment could thus support breathing mechanics and tissue repair, delaying fibrosis.

By day 3 post-bleomycin, immune cell infiltration and fibroblast recruitment continue [32]. NPPS-PE may impact fibroblasts before fibrotic foci form, reducing mechanosensing stimuli, inducing apoptosis, and lowering collagen synthesis [16,17]. The alveolar integration of NPPS-PE could regulate fibroblast activity and matrix deposition. Surfactant-induced calcium signaling via the IP3 pathway may contribute to fibroblast apoptosis [14,16]. This mechanism could help preserve alveolar structure, as seen in histological analysis.

By day 7, fibrosis begins, with fibroblasts disrupting gas exchange and type II alveolar cell apoptosis occurring [31]. Surfactant recycling is impaired [35], and basement membrane integrity is lost, promoting fibrosis. NPPS-PE incorporation may be reduced at this stage, suggesting higher efficacy during inflammation. Overall, NPPS-PE may modulate inflammation, stabilize surfactant function, and protect basement membrane integrity, limiting fibrosis progression. Its action contrasts with current antifibrotic therapies like nintedanib and pirfenidone, which lack surface activity. NPPS-PE represents a dual-action therapy, combining biophysical and biological effects, particularly effective during early fibrotic events (Figure 9).

Day 0—Early inflammatory phase: Bleomycin induces epithelial damage and promotes the release of pro-inflammatory cytokines (IL-6, IL-1β, TNF-α). NPPS-PE particles integrate into the epithelial monolayer, suggesting modulation of the inflammatory response and protection of the basement membrane.Day 3—Onset of fibrogenesis: Epithelial disruption progresses (1). Loss of basement membrane integrity enables TGF-β signaling from macrophages, which recruits fibroblasts (2). NPPS-PE may still modulate fibroblast activity by reducing proliferation and collagen expression (3).Day 7—Fibrotic stage: Persistent epithelial injury, apoptosis of type II alveolar cells, and surfactant dysfunction promote the formation of fibrotic foci. NPPS-PE is no longer effective, and structural damage leads to impaired gas exchange.

One limitation of this study is the use of racemic PE, which contains both enantiomers. Given the potential biological and biophysical differences between lipid enantiomers, it would be highly relevant to investigate whether the enantiomers of phosphatidylethanolamine exert differential effects on the biological responses observed. Such studies could help clarify whether the use of enantiomerically pure phosphatidylethanolamine enhances the efficacy or specificity of its cellular effects compared to racemic mixtures.

The NPPS-PE mixture holds potential as a therapeutic option, like other commercial surfactants, for treating surfactant-related diseases in neonates, especially in our country, where the availability of standard exogenous surfactants is often limited due to cost and supply-chain constraints, making locally produced or adapted formulations such as NPPS-PE a potentially impactful therapeutic option.

The attenuation of lung fibrosis in the bleomycin model is only seen in the inflammatory phase, suggesting that it could be of benefit in post inflammatory lung fibrosis. We are currently investigating the effect of NPPPS-PE in an animal model of LPS-induced pulmonary fibrosis.

## 4. Materials and Methods

### 4.1. Preparation, Characterization, and Biophysical Evaluation of NPPS-PE

Porcine lungs were obtained from the local abattoir according to specific criteria: without larynx, heart, or tissue lesions. Bronchoalveolar lavage was performed with 1 L of 0.9% saline solution. The collected bronchoalveolar fluid was centrifuged at 3500 rpm for 5 min to remove cellular debris, followed by a second centrifugation at 9000 rpm to obtain a pellet with the large surfactant complexes. The pellet was then processed through a series of solvent systems consisting of methanol:chloroform:water (2:1:0.8). After filtration and separation of the chloroform phase, the pellet was treated with acetone to increase the volume 20× for lipid reconstitution and evaporated under nitrogen gas. The resulting pellet was subjected to evaporation to remove solvent traces and later resuspended in a solution of NaCl 100 mM-CaCl_2_ 5 mM. Finally, the mixture was sterilized using a flash autoclave at 120 °C for 10 min to ensure a sterile and homogeneous product. Total phospholipids in the NPPS were quantified by the Rouser method, which uses phosphorus analysis to measure phospholipid content. Phospholipid classes were analyzed by thin-layer chromatography.

### 4.2. Evaluation of NPPS-PE Monolayer Properties Using the Langmuir Film Balance

Monolayer experiments were performed at 25 °C using a Langmuir–Blodgett balance (102M micro Film Balance; NIMA Technologies, Coventry, UK) as previously described [16]. The aqueous subphase contained NaCl 100 mM-CaCl_2_ 5 mM at 25 °C. A 1 mg/mL concentrated solution of the organic lipid extract of NPPS or PE dissolved in chloroform:methanol (1:2) was used for monolayer experiments. The monolayers were formed by spreading 10 µL of a concentrated solution of the organic lipid extract of NPPS-PE at the air–water. After evaporation of the organic solvent, the monolayer was allowed to stabilize for 10 min. Monolayers were then cyclically compressed five times.

### 4.3. Observation of Rhodamine-DOPE-Labeled NPPS-PE Films by Epifluorescence Microscopy

NPPS and PE samples were labeled with 0.1% rhodamine-DOPE (Avanti Polar Lipids, Alabaster, AL, USA) suspended in chloroform:methanol (1:2). Subsequently, 20 µL of the rhodamine-DOPE-labeled NPPS-PE suspension was spread onto the air–liquid interface of the Langmuir film balance, and the organic solvent was allowed to evaporate for 10 min. The interfacial film was then transferred onto a glass coverslip during compression at a constant rate of 25 cm^2^/min using the Langmuir–Blodgett technique with a continuously varying surface pressures (COVASP) mode [36]. The transferred films were observed using an epifluorescence Leica DM 4000B microscope (Leica Microsystems, Wetzlar, Germany) equipped with an ORCA R2 10,600 camera (Hamamatsu Photonics K.K., Shizuoka, Japan).

### 4.4. Surface Tension Measurement with a Captive Bubble Surfactometer

The interfacial activity of NPPS samples in the absence and presence of PE suspended in NaCl 100 mM-CaCl_2_ 5 mM was evaluated in a captive bubble surfactometer at 37 °C as described in [22]. The surfactometer chamber contained 5 mM Tris-HCl, pH 7, buffer with 150 mM NaCl, and 10% sucrose. After a small air bubble (0.035–0.040 cm^3^) had formed, approximately 200 nL of surfactant (20 mg/mL) was deposited beneath the bubble surface with a transparent capillary. Then, the change in surface tension (γ) was monitored for 5 min from the changes in the shape of the bubble as recorded in a digital camera. The chamber was sealed, and the bubble was rapidly expanded (within 1 s) to a volume of 0.15 cm^3^ to record the post-expansion adsorption. Five minutes after expansion, quasi-static cycles began, in which the bubble was initially reduced in size (to 20% of its previous volume) and then gradually enlarged. A 1 min delay was maintained between each of the four quasi-static cycles, and an additional 1 min delay was added before dynamic cycles started, in which the bubble was continuously compressed and expanded at a rate of 20 cycles per min.

### 4.5. Isolation and Maintenance of Normal Human Lung Fibroblasts

The isolation of primary NHLFs was performed in our laboratory as previously described [16]. Briefly, NHLFs were obtained from lung tissue donated by patients who were diagnosed as brain dead. Family consent was obtained before the procedure, and these donors had no history of lung disease, smoking, allergy, or comorbidities. The Ethics and Research Committees of the Hospital General de Puebla and the Benemérita Universidad Autónoma de Puebla, Faculty of Medicine, reviewed and approved the study protocol with a registration number (SIEP/C.I/153/2022). The lung piece was obtained from the right lingula with a dimension of 5 × 2 cm in thickness, which is preserved in 30 mL of Ham’s F-12 medium without fetal bovine serum. A portion was processed for histopathology. The other portion was minced and incubated for 20 min in 1X trypsin-EDTA solution and serum-free F-12 medium. The digested tissue was gently triturated with a 10 mL pipette, and the dissociated cells were filtered through a mesh filter. Filtrate was centrifuged at 200× *g* for 10 min using an Centrifuge model 5804R (Eppendorf SE, Hamburg, Germany), and the pellet was resuspended in F-12 medium containing 10% fetal bovine serum (FBS). Cells were cultured in T-25 flasks and grown to 75% confluence in F-12 medium supplemented with 10% FBS, 100 U/mL penicillin, and 100 μg/mL streptomycin at 37 °C in 95% O_2_, 5% CO_2_ saturated humidity. Only cells that were derived from lungs with a normal histology, as determined by histopathological examination, were included in this study. Fibroblasts from passages 5 to 10 were plated on coverslips in petri dishes, allowed to adhere for 24 h, and then incubated in serum-free medium for 48 h.

### 4.6. Cell Proliferation Assay

A Neubauer chamber was used to generate a cell concentration curve with values of 62,500, 46,875, 31,250, 15,625, and 7813 cells/cm^2^. These cells were seeded in a 96-well plate format. For the experimental conditions, a concentration of 15,625 cells/cm^2^ was used, and all conditions were seeded in triplicate. After seeding the cells and incubating them overnight, the medium was removed, and the experimental conditions were applied. The experimental setup included the following conditions: cells treated with transforming growth factor-beta (TGF-β) at 1 ng/mL as an inducer of cell proliferation, cells treated with NPPS (500 µg/mL), cells treated with PE (100 µg/mL), and cells treated with NPPS-PE (500 µg/mL and 100 µg/mL, respectively). Cell proliferation was measured at 24 and 48 h using the WST-1 kit according to the manufacturer’s instructions. Absorbance readings were taken at 550 nm.

### 4.7. Evaluation of Apoptosis in NHLF Treated with NPPS-PE by Flow Cytometry

NHLF cells were seeded in 24-well plates at a density of 50,000 cells/mL. Apoptosis was assessed using Annexin-V and propidium iodide according to the manufacturer’s recommendations. Apoptosis measurement by flow cytometry was performed using a CytoFlex instrument (Beckman Coulter, Indianapolis, IN, USA), and microscopic analysis was performed using a ZOE™ Fluorescent Cell Imager (Bio-Rad Laboratories, Hercules, CA, USA). NHLF cells were treated with medium as a control, NPPS (500 µg/mL), PE (100 µg/mL), and NPPS-PE (500–100 µg/mL). Measurements were performed in triplicate at 24 and 48 h.

### 4.8. Determination of COL1A1 Expression by RT-qPCR

Total RNA extraction was performed using TRIzol (Ambion by Life Technologies, Austin, TX, USA; Cat. No. 15596-026), according to the manufacturer’s instructions. NHLF cells were seeded in 6-well plates at a density of 300,000 cells per well, and the experimental groups were as follows: control cells cultured in Ham-F12 medium without fetal bovine serum, NPPS-treated cells (500 µg/mL), PE-treated cells (100 µg/mL), and NPPS-PE-treated cells (500–100 µg/mL). The incubation time for the experimental conditions was 24 and 48 h. The extracted RNA was quantified using the NanoDrop 100 spectrophotometer (Thermo Fisher Scientific, Waltham, MA, USA) and stored at −70 °C. RNA was converted to cDNA using a thermal cycler and the H Minus First Strand cDNA Synthesis Kit (Thermo Scientific, USA). RT-qPCR was performed using *GAPDH* (Hs02786624_g1) as the reference gene and *COL1A1* (Hs00164004_m1) as the target gene. The PCR conditions were set as 2 min at 94 °C, followed by 40 cycles of 15 s at 95 °C and 1 min at 60 °C, finishing with an indefinite cooling loop. Each experimental sample was loaded in triplicate, along with a triplicate of the *GAPDH* gene. Samples were processed on a Rotor-Gene Q thermal cycler (Qiagen, Hilden, Germany).

### 4.9. Determination of the Effect of NPPS-PE in C57BL/6 Mice Treated with Bleomycin

This experimental protocol was approved by the CICUAL (Institutional Animal Care and Use Committee) of the Benemerita Universidad Autonoma de Puebla. All animals were housed under controlled environmental conditions and a 12 h light/dark cycle. Animals were maintained in ventilated cages with standard bedding material and had ad libitum access to food and water. All housing and husbandry conditions were in accordance with international guidelines to ensure animal welfare and minimize stress. The bleomycin model was established using C57BL/6 mice, weighing 30–40 g and six weeks old. The sample size was calculated considering the response seen in a previous study [15]. Mice were randomly allocated to each group using the random allocation rule with the R package randomizeR, and confounders were not controlled. The provider of the treatment was aware of the group allocation and was not in charge of the measurements. The animals were anesthetized with a mixture of xylazine/ketamine (20 and 100 mg/mL, respectively) at a dose of 0.2 mL per 100 g, and bleomycin instillation was performed intratracheally at a dose of 3U/kg in 30 µL [16]. Treatments were administered through nebulization, with NPPS-PE 400–80 mg/kg diluted to 1.5 mL with saline solution. Treatments commenced on days 0, 3, and 7 (Figure 9). The animals were euthanized on day 20 under anesthesia. No animals were excluded. The right lung was used for hydroxyproline quantification, while the left lung was subjected to H&E and Masson’s Trichrome staining, with analysis performed using Orbit Image Analysis machine learning software version 3.64.

### 4.10. Pulmonary Surfactant Deficiency Model in Animal Models of Surfactant Deficiency Treated with NPPS-PE

A total of 6 Sprague-Dawley rats (weighing 250–300 g) and 6 guinea pigs (*Cavia porcellus*) were obtained and divided into two groups according to species: a control group with the vehicle and a group treated with NPPS-PE (100 mg/kg of body weight) from a NPPS-PE preparation (25 and 5 mg, respectively). No criteria for including or excluding animals were used.

The treatments were suspended in NaCl 100 mM-CaCl_2_ 5 mM. The animals were anesthetized with a mixture of xylazine/ketamine (20 and 100 mg/mL, respectively) at a dose of 0.2 mL per 100 g. They were then immobilized and placed in a supine position with the jaw retracted. An incision was made at the level of the trachea, which was then localized and cannulated. The animals were then placed on a positive pressure ventilator Harvard Small Animal Mouse Ventilator, Model 687, 55-0001 (Harvard Apparatus, Holliston, MA, USA) delivering O_2_ at a flow rate of 2 L/min. Rats were ventilated at a frequency of 90 breaths per min, guinea pigs were ventilated at 100 breaths per min, with an O_2_ volume of 2.5 mL per breath. Three lung lavages were performed over a 30 min period, alternating with mechanical ventilation. Each lavage was performed with 0.9% saline solution at a rate of 35 mL/kg at 37 °C. After the lung lavage and induction of pulmonary surfactant deficiency, the appropriate treatment was administered. The guinea pigs’ right atrium was punctured to obtain blood samples of 300 µL, which were analyzed using a Blood gas analyzer GEM Premier 3500 (Instrumentation Laboratory, Bedford, MA, USA) at times 0, 30, 60, 90, and 120 min. During the experimental phase, oxygen saturation (SpO2) was monitored using a pediatric pulse oximeter placed on the left paw.

### 4.11. Statistical Analysis

The results are expressed as mean ± standard deviation of experiments performed in triplicate. Data analysis of the biophysical assessment was performed using Prism v.9.5.0 software (GraphPad, San Diego, CA, USA), while the remaining data were processed using R 4.4.1. Statistical inferential analysis was performed with ANOVA using a significance level of *p* < 0.05, and a bilateral Dunnett’s post-hoc test was conducted.

## 5. Conclusions

In conclusion, we have demonstrated that the NPPS-PE mixture exhibits significant potential as a novel pulmonary surfactant with both biophysical and therapeutic benefits. The composition of NPPS enriched with PE showed synergistic effects in improving surfactant activity and optimizing oxygenation in an animal model of surfactant deficiency. Furthermore, NPPS-PE demonstrated antifibrotic properties in vitro, reducing collagen deposition and promoting apoptosis in fibroblasts, thereby offering a promising approach for diseases associated with pulmonary fibrosis. The biochemical and biophysical properties of NPPS-PE, comparable to commercial surfactants, support its use as an alternative treatment for respiratory conditions, particularly in neonates. Although further studies are needed to investigate its enantiomeric composition and the potential blockage of TGF-β pathways, our findings suggest that NPPS-PE could serve as both a prophylactic and therapeutic option for lung diseases, including those leading to fibrosis. Given the promising results observed in this study, NPPS-PE should be further explored in clinical trials, particularly for conditions like diffuse interstitial pneumonia in children, where effective treatments are currently lacking. Ultimately, NPPS-PE presents a valuable opportunity for advancing lung disease therapies, offering hope for improved management and outcomes in pediatric and neonatal patients.

## 6. Patents

Patent application number for the organic synthesis of 1,2-dipalmitoyl-*rac*-glycero-3-phosphatidylethanolamine: MX/E/2024/071586.

Patent application number for the NPPS-PE: MX/E/2024/083492

## Figures and Tables

**Figure 1 ijms-26-07132-f001:**
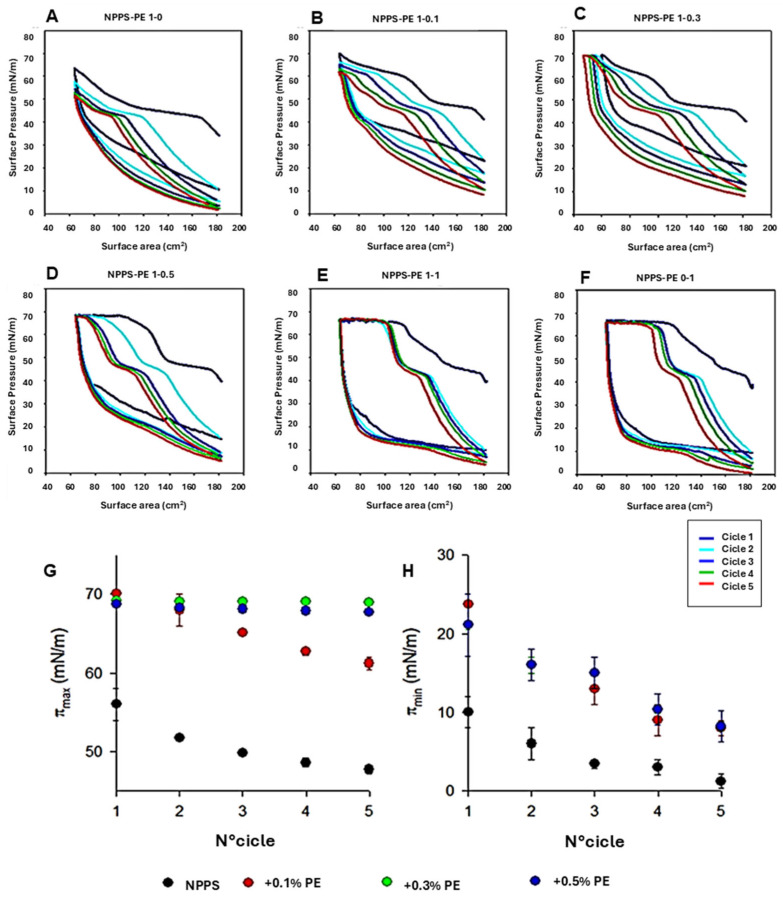
Effect of PE on NPPS compression/expansion cycles. (**A**–**F**) π-area isotherms obtained from five cycles for pure NPPS monolayers (**A**), NPPS-PE blends with molar ratios 1:0.1 (**B**), 1:0.3 (**C**), 1:0.5 (**D**), and 1:1 (**E**), and pure PE (**F**). (**G**,**H**) Effect of PE on the maximum (πmax) (**G**) and minimum (πmax) (**H**) surface pressure reached by the isotherms of NPPS isolated from porcine lung lavages during compression cycles.

**Figure 2 ijms-26-07132-f002:**
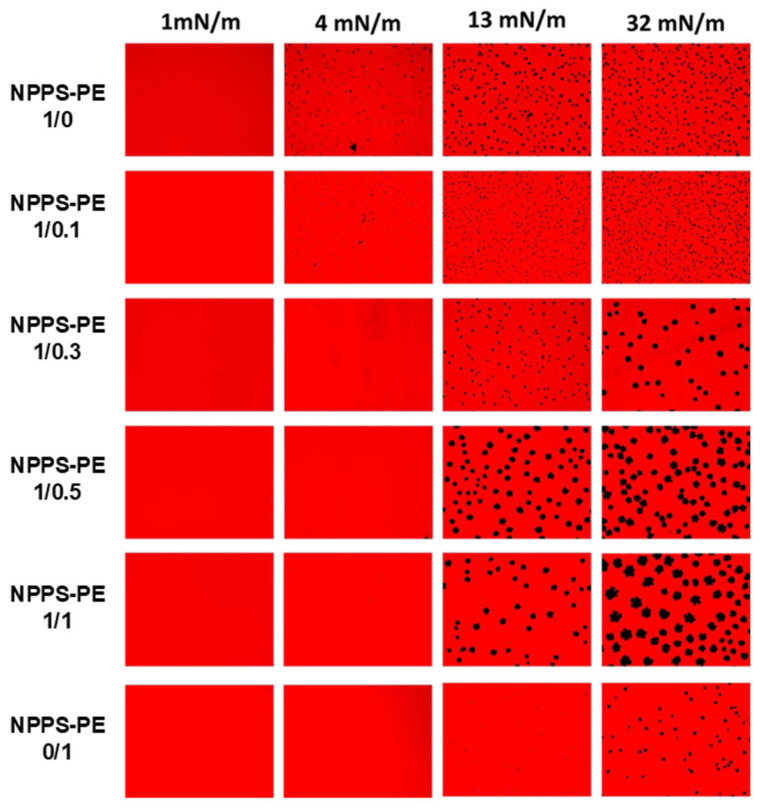
Micrographs obtained from the transfer of NPPS-PE (1/0, 1/0.1, 1/0.3, 1/0.5, 1/1, and 0/1) stained with Rho-DPPE, experiments performed on the Langmuir balance. Images of NPPS were obtained alone and in the presence of PE. The transfers of the interfacial films were performed on glass slides and observation under an epifluorescence microscope. Monolayers in a homogeneous expanded liquid (LE) phase are shown at 1 mN/m. Further compression of the monolayer resulted in the nucleation and growth of domains in the crystal phase (CP) with gradual incorporation of PE into NPPS.

**Figure 3 ijms-26-07132-f003:**
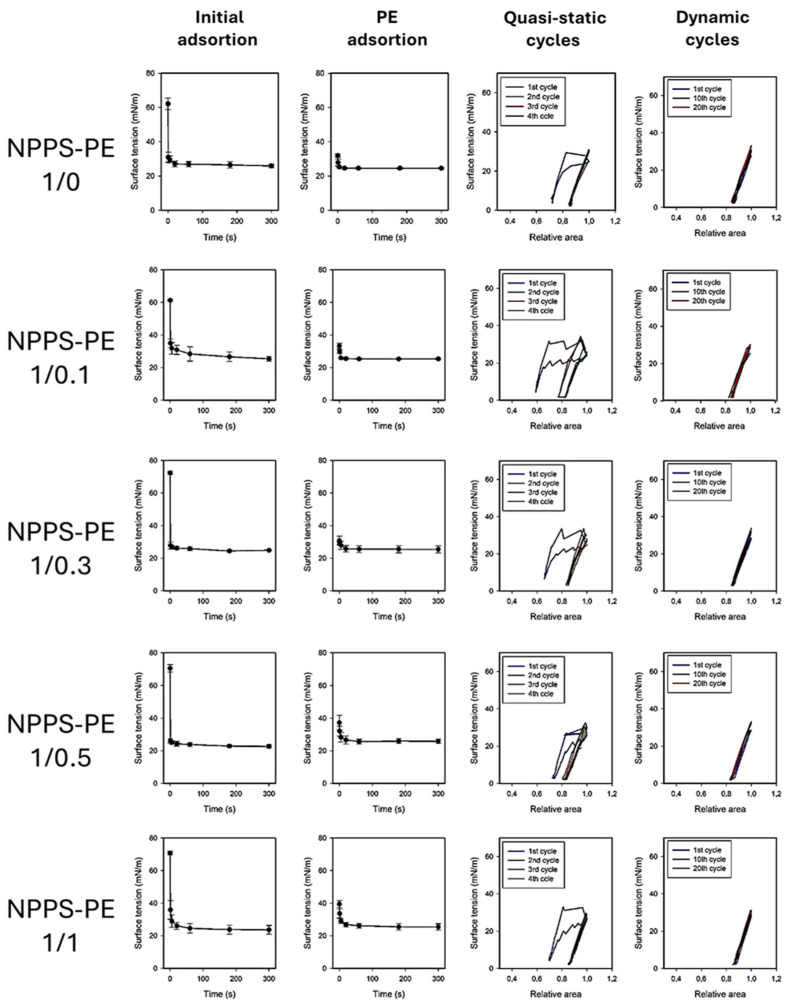
Surface tension evolution of PE-enriched NPPS at different molar ratios (1/0, 1/0.1, 1/0.3, 1/0.5, 1/1, and 0/1). The first two columns show the change in surface tension as a function of time during initial adsorption and post-expansion. The last two columns represent the quasi-static and dynamic compression–expansion cycles. A captive bubble surfactometer was used to measure, with an average of three experiments, and the standard deviation represented by the error bars.

**Figure 4 ijms-26-07132-f004:**
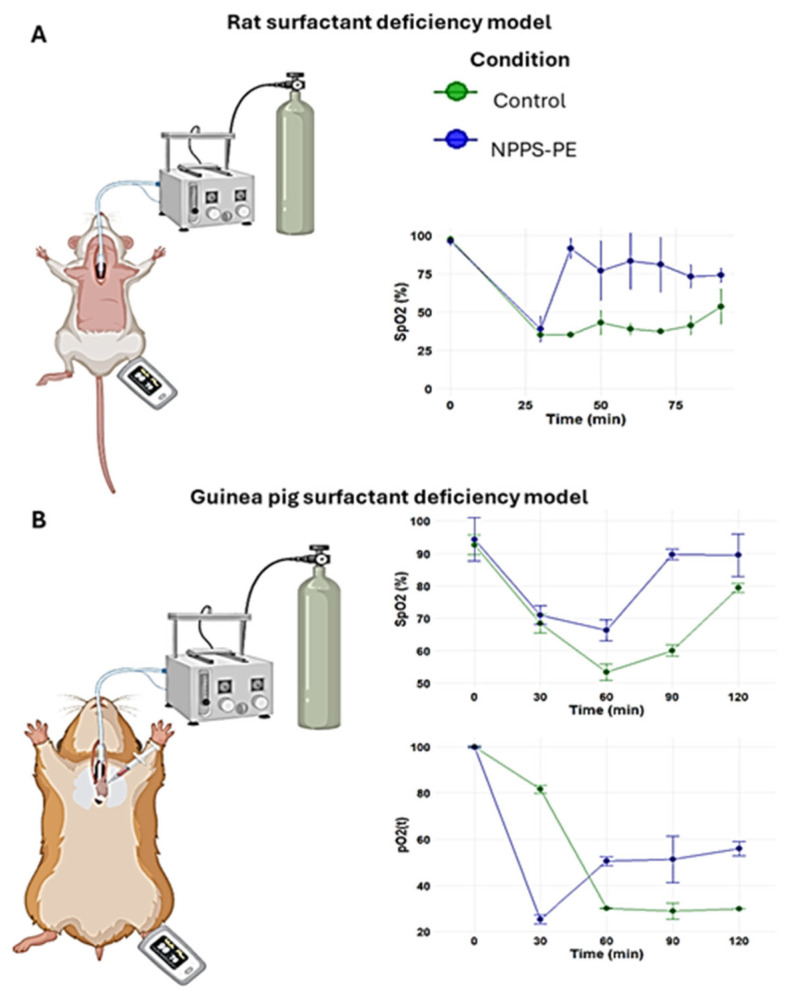
Surfactant deficiency models in animals. (**A**) Oxygen saturation (SpO_2_) was monitored using pulse oximetry in rats, while (**B**) partial oxygen pressure (PaO_2_) and SpO_2_ were quantified in blood samples from the right ventricle in guinea pigs. In both models, the groups were the control group and the group treated with NPPS-PE at multiple time points. The results showed that treatment with NPPS-PE improved respiratory parameters over time in both models.

**Figure 5 ijms-26-07132-f005:**
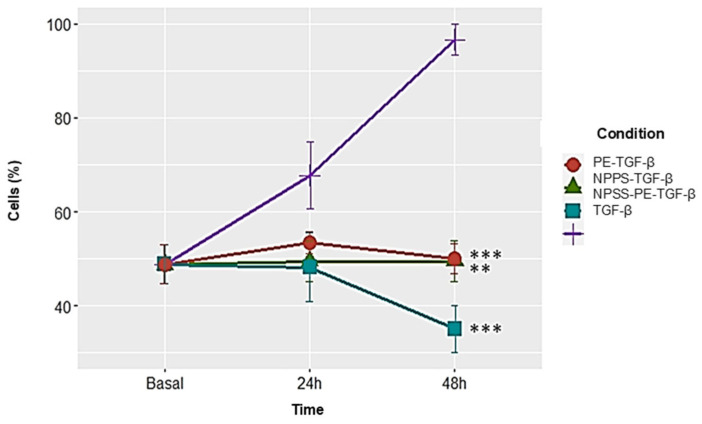
NPPS enriched with PE decreases cell proliferation in NHLF. The growth rate was assessed using the WST-1 assay in three NHLF cell lines derived from three different donors at 24 and 48 h in cells treated with TGF-β as a growth and proliferation inducer. The data presented are the average of three replicates from three independent experiments. Statistical significance: *p* < 0.001 ** and *p* < 0.0001 ***.

**Figure 6 ijms-26-07132-f006:**
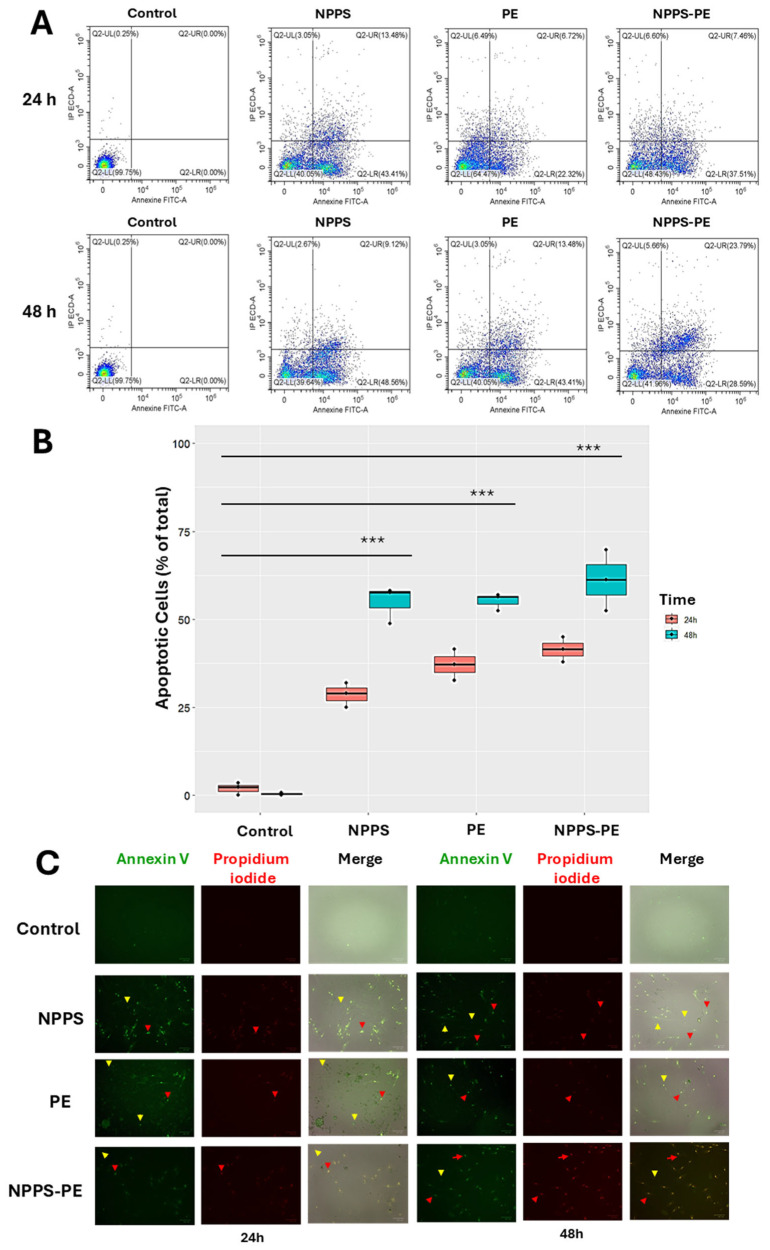
Induction of apoptosis with NPPS-PE evaluated by flow cytometry at 24 and 48 h. The analysis included the following aspects: (**A**) A graphical representation is provided showing the number of cells in early and late apoptosis in the Q2-LR and Q2-UR panels, respectively. (**B**) Box plots are shown representing triplicate experiments across three different cell lines and the sum of the quadrants representing apoptosis. Statistical analysis was performed with a significant level of *p* < 0.05. (**C**) Representative microphotographs of stained cells using the same fluorophores employed in the cytometry experiments (annexin V and propidium iodide). Yellow arrows indicate the presence of early apoptosis, and red arrows indicate late apoptosis. *p* < 0.0001 ***.

**Figure 7 ijms-26-07132-f007:**
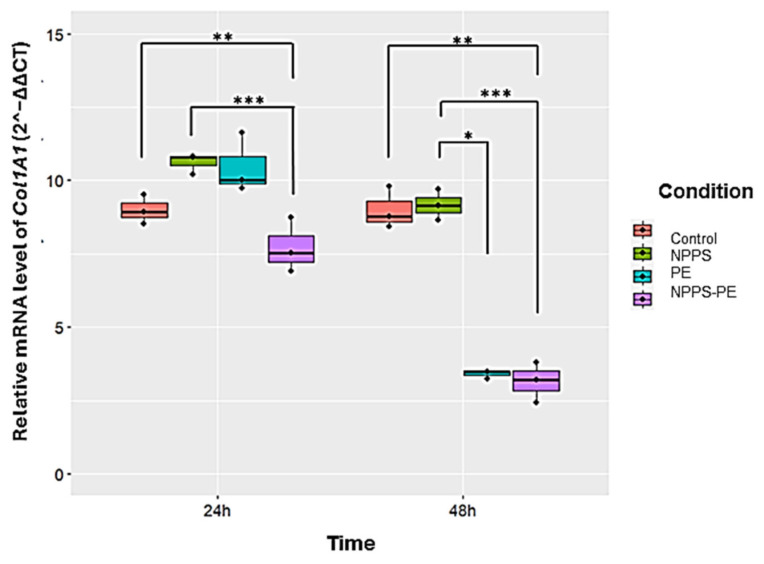
The effect of NPPS enriched with PE on normal human lung fibroblasts (NHLFs) collagen expression (Col1A1) was determined by RT-qPCR. The graphical results reveal that at 24 and 48 h, the combination of NPPS-PE leads to a significant decrease in Col1A1 expression in NHLF. It is important to note that when NPPS was enriched with PE, its inhibitory effect on Col1A1 expression was notably enhanced. *p* < 0.01 *, *p* < 0.001 ** and *p* < 0.0001 ***.

**Figure 8 ijms-26-07132-f008:**
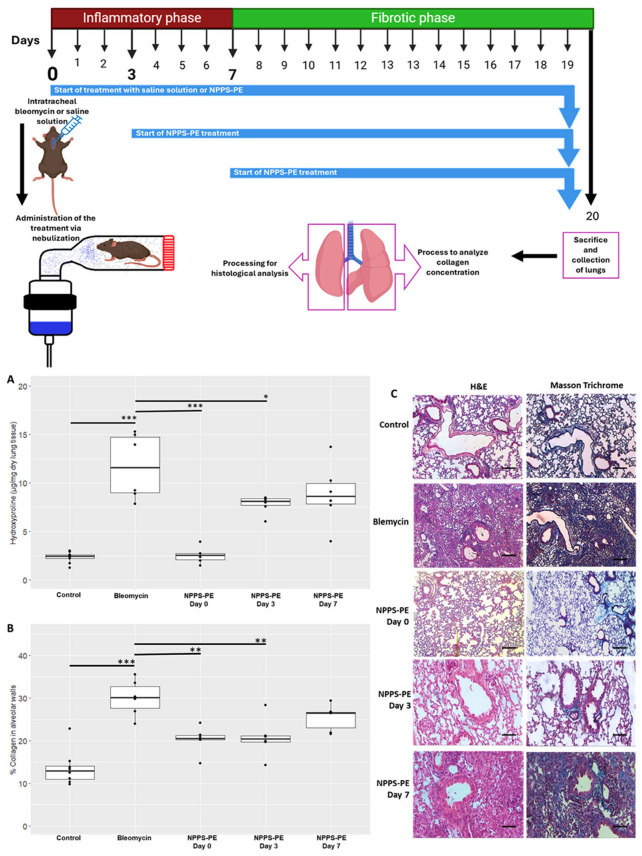
Effect of the timing of NPPS-PE treatment initiation on disease progression in a bleomycin-induced pulmonary fibrosis model. (**A**) Experimental Timeline: Pulmonary fibrosis was induced by a single dose of bleomycin (day 0). Mice were treated with either saline solution (control) or NPPS-PE at three different time points: at the onset of injury (day 0), during the inflammatory phase (day 3), and in the early fibrotic phase (day 7). Lungs were collected on day 20 for histological and biochemical analysis. The treatment was administered by inhalation, placing the mice in a containment chamber. (**B**) Collagen Deposition: Hydroxyproline levels, a marker of collagen deposition, significantly increased following bleomycin administration compared to controls. NPPS-PE treatment significantly reduced these levels in the groups treated on days 0 and 3 (* *p* < 0.05, ** *p* < 0.01, and *** *p* < 0.001). The percentage of collagen in alveolar walls, assessed by histological staining, showed an increase in bleomycin-treated mice, which was partially reversed by NPPS-PE treatment on days 0 and 3 (* *p* < 0.05, ** *p* < 0.01, and *** *p* < 0.001). (**C**) Histological Images: Lung sections stained with Masson’s Trichrome revealed collagen accumulation (blue areas) following bleomycin treatment; reduction is only seen when NPPS-PE is administered during the inflammatory phase (0 and 3 days). Bar represents 50 µM.

**Figure 9 ijms-26-07132-f009:**
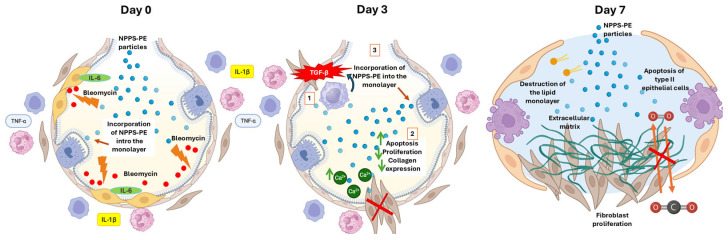
Proposed mechanism of action of NPPS-PE in bleomycin-induced pulmonary fibrosis.

## Data Availability

The data supporting the results of the current study are available from the corresponding author upon reasonable request.

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
