# Peer review of "Modulation of Pulmonary Fibrosis by Pulmonary Surfactant-Associated Phosphatidylethanolamine In Vitro and In Vivo"

_ijms, 2025, doi:10.3390/ijms26157132_

Round 1

Reviewer 1 Report

Comments and Suggestions for Authors

The study discusses the therapeutic efficacy of a natural porcine pulmonary surfactant (NPPS) enriched with phosphatidylethanolamine (PE) for the treatment of pulmonary fibrosis. The researchers reported in vivo and in vitro evidence that NPPS-PE inhibits fibroblast proliferation, promotes their apoptosis, and inhibits type I collagen (Col1A1) expression levels. The authors also illustrated how NPPS-PE enhances surfactant activity and oxygenation in surfactant-deficient experimental models at the same time, inducing a remarkable inhibition of collagen deposition in a model of bleomycin-induced pulmonary fibrosis. It is finally proposed that NPPS-PE might be applied both for prophylactic and therapeutic administration in pulmonary diseases, particularly in children.

This study is of specific significance as it integrates biophysical experiments, cell studies, and animal models to introduce a novel strategy in the prevention and treatment of pulmonary fibrosis. Since the disease is a chronic and progressive disorder for which there is no curative therapy, the application of a modified surfactant like NPPS-PE can change the therapeutic strategy dramatically. The article also increases understanding on the role of phospholipids in the pathophysiological processes of fibrosis and suggests possible uses in the practice of pediatric pulmonology.

The manuscript needs several major revisions.

  1. The abstract is dense with information, which can distract the reader from the central message. Simplifying the technical information regarding the source of the surfactant and highlighting more clearly the central concept: that NPPS-PE has therapeutic potential for pulmonary fibrosis would be beneficial. A shorter version with a focus on translating the results into future clinical applications would make the article easier to read for the non-expert reader.
  2. The discussion section resolves numerous points of difference, but the prose is extensive and there is some repetition of ideas. A more condensed structure, whereby findings in both in vitro and in vivo models are obviously connected to biophysical findings, would perhaps render the document more readable. Meanwhile, it would be beneficial to focus more on mechanisms of action of NPPS-PE and to describe how these differ from current or experimental treatment methodologies.
  3. While NPPS-PE is characterized by higher efficacy during the inflammatory phase (day 0 or 3), the significance of this finding warrants more attention. It could be helpful to recommend early administration protocols with targeting or compare with other therapies (e.g., TGF-β inhibitors) for increasing the robustness of the findings.
  4. While the authors propose the use of NPPS-PE in neonatal and pediatric patients, its applicability to current clinical practice is modest. Inclusion in the discussion section of citations for successful, as well as failed, attempts at utilizing modified surfactants in pediatric models and patients would enhance the translational potential of the study.
  5. Certain sections, particularly in the "Discussion" portion, are characterized by lengthy sentences with dense and compact wording. These can be paraphrased into shorter paragraphs and simpler sentence structures to make the article more readable. One example is the explanation of Figure 7 showing the stages of fibrosis and NPPS-PE's impact; it would be clearer if presented in bullet points or short sentences.
  6. The mention of racemic PE usage is relevant; however, its implications in pharmacodynamics are somewhat minimal. It would be beneficial to explain how the selection of numerous enantiomers might influence therapy regimens. It would also be beneficial to include a discussion on the problems in manufacturing, sustaining stability, and administering these preparations in human patients.
Comments on the Quality of English Language

The manuscript is generally well written, with only minor grammatical and syntactical issues.

Author Response

Reviewer 1

Comments 1: The abstract is dense with information, which can distract the reader from the central message. Simplifying the technical information regarding the source of the surfactant and highlighting more clearly the central concept: that NPPS-PE has therapeutic potential for pulmonary fibrosis would be beneficial. A shorter version with a focus on translating the results into future clinical applications would make the article easier to read for the non-expert reader.

Response 1: Thank you for your thoughtful comment. As recommended, we revised the abstract to simplify technical details while maintaining key information. We clarified that the surfactant was evaluated biophysically, and the improvements observed in surface pressure and stability are now briefly mentioned. We also corrected the description of the in vitro experiments to specify that TGF-β was only used in the proliferation assays. The revised abstract focuses more clearly on the therapeutic potential of NPPS-PE and its relevance for pulmonary fibrosis. We appreciate your guidance in improving the clarity and accessibility of our manuscript.

Comments 2: The discussion section resolves numerous points of difference, but the prose is extensive and there is some repetition of ideas. A more condensed structure, whereby findings in both in vitro and in vivo models are obviously connected to biophysical findings, would perhaps render the document more readable. Meanwhile, it would be beneficial to focus more on mechanisms of action of NPPS-PE and to describe how these differ from current or experimental treatment methodologies.

Response 2: In response, we have revised the Discussion section to reduce redundancy and improve clarity. We reorganized the content to create a more condensed and integrated structure, explicitly connecting the biophysical findings with both the in vitro and in vivo results. We also expanded our interpretation of the potential mechanisms of action of NPPS-PE, including its influence on fibroblast activity and surfactant function. Finally, we included a comparison with current antifibrotic treatments, highlighting the unique dual-action profile of NPPS-PE. We hope that these changes improve the readability and focus of the Discussion section as per your recommendations.

Comments 3: While NPPS-PE is characterized by higher efficacy during the inflammatory phase (day 0 or 3), the significance of this finding warrants more attention. It could be helpful to recommend early administration protocols with targeting or compare with other therapies (e.g., TGF-β inhibitors) for increasing the robustness of the findings.

Response 3: We appreciate this important suggestion. While our current findings highlight the enhanced efficacy of NPPS-PE during the early inflammatory phase (day 0 to 3), we agree that this aspect warrants further investigation. In response, we are currently developing a more clinically relevant model of acute lung injury based on systemic inflammation, such as that observed in sepsis. Sepsis represents a systemic inflammatory response to infection that often results in acute lung injury (ALI) or acute respiratory distress syndrome (ARDS), providing a translationally meaningful context in which to evaluate the therapeutic potential of NPPS-PE. This ongoing work will allow us to assess its effects in a more dynamic and heterogeneous inflammatory environment. We hope this information demonstrates our commitment to strengthening the translational relevance and robustness of our findings. We rephrased the paragraph, narrowing its potential clinical use to post inflammatory lung fibrosis.

Comments 4: While the authors propose the use of NPPS-PE in neonatal and pediatric patients, its applicability to current clinical practice is modest. Inclusion in the discussion section of citations for successful, as well as failed, attempts at utilizing modified surfactants in pediatric models and patients would enhance the translational potential of the study.

Response 4: Thank you for this important observation. While we acknowledge that the clinical applicability of modified surfactants remains a challenge in some contexts, we respectfully argue that in low- and middle-income countries (LMICs) such as Mexico, the need for affordable, effective, and accessible pulmonary surfactant therapies remains a pressing clinical priority. In these settings, the availability of standard exogenous surfactants is often limited due to cost and supply-chain constraints, making locally produced or adapted formulations such as NPPS-PE a potentially impactful therapeutic option. Furthermore, our focus on natural porcine-derived surfactant enriched with PE aims not only to enhance biophysical and antifibrotic properties but also to ensure compatibility with scalable production methods feasible in public health institutions. While acknowledging that the translation to widespread neonatal and paediatric use requires further validation, we believe the development of such alternatives is crucial to improving outcomes in vulnerable populations, especially in countries where healthcare resources are limited.

In response to your suggestion, we have modified the potential clinical use to refer to post-inflammatory lung fibrosis and added the locally potential benefit concerning its use in neonates.

Comments 5: Certain sections, particularly in the "Discussion" portion, are characterized by lengthy sentences with dense and compact wording. These can be paraphrased into shorter paragraphs and simpler sentence structures to make the article more readable. One example is the explanation of Figure 7 showing the stages of fibrosis and NPPS-PE's impact; it would be clearer if presented in bullet points or short sentences.

Response 5: We appreciate your observation regarding the dense sentence structure in parts of the Discussion. In response, we have revised several sections to improve readability by paraphrasing lengthy sentences into shorter paragraphs with simpler syntax. Specifically, the description related to Figure 9—used to illustrate the stages of fibrosis and the proposed mechanism of NPPS-PE—has been reformatted using bullet points and concise sentences. We hope these changes enhance clarity and make the manuscript more accessible to a broader audience.

Comments 6: The mention of racemic PE usage is relevant; however, its implications in pharmacodynamics are somewhat minimal. It would be beneficial to explain how the selection of numerous enantiomers might influence therapy regimens. It would also be beneficial to include a discussion on the problems in manufacturing, sustaining stability, and administering these preparations in human patients.

Response 6: We appreciate the reviewer’s insightful comments regarding the use of racemic phosphatidylethanolamine (PE) and its implications for pharmacodynamics. We recognize that the presence of multiple enantiomers may influence therapeutic efficacy and regimen optimization. Therefore, we are actively working on the organic synthesis of an enantiomerically pure PE mixture to better understand and potentially enhance the therapeutic profile of NPPS-PE. Additionally, we acknowledge the challenges related to manufacturing, stability, and administration of these surfactant preparations in clinical settings. These considerations are being addressed in our ongoing development process, with a focus on ensuring scalable synthesis methods, maintaining formulation stability, and optimizing delivery routes for human patients. We have added a discussion reflecting these points in the revised manuscript to highlight the future directions and translational challenges involved.

Reviewer 2 Report

Comments and Suggestions for Authors

This study explored the role of surfactant enriched with 1,2-di-palmitoyl-rac-glycero-3-phosphatidylethanolamine (PE) in lung fibrosis. They developed a natural porcine surfactant enriched with PE (NPPS-PE) obtained by organic synthesis and evaluated its biophysical properties, which is their innovation point. However, there are also many points that need to be modified.

Major points:

1.The author only explored proliferation and apoptosis at the cellular level. This study lacktherelated molecular mechanisms and signaling pathways such as NF-κB, TGF-β and other pathways.

2.NPPS-PE is extracted from pigs. Does NPPS-PE have a rejection effect? Can all species be used?Please provide a specific description of its relevant scope of adaptation and adverse reactions

Miner points:

1.The image pixels are too low;

2.Figure 3 shows only one cell proliferation, and the article also mentions that NHIF has three cell lines. Is this the result of using one cell? Three types of cells, shouldn't they be three small pictures? There are various methods for detecting proliferation, such as CCK-8, EDU, etc; Suggest adding different methods for detection and multi-party verification;

3.Figure 4 shows two detection methods, flow cytometry and immunofluorescence. Figure legend only shows flow cytometry.

4.In Figure 5, RT-PCR was used to verify the decrease in expression of Col1A1. Why not use protein level validation such as WB?

5.The HE and Masson staining in Figure 6 showed the most significant effect after 3 days of NPPS-PE administration, which differed greatly from the results in Figure 6B.

Author Response

Reviewer 2

Major points:

Comments 1. The author only explored proliferation and apoptosis at the cellular level. This study lack the related molecular mechanisms and signaling pathways such as NF-κB, TGF-β and other pathways.

Response 1: We sincerely thank the reviewer for this insightful comment. We agree that exploring the underlying molecular mechanisms and signaling pathways such as NF-κB and TGF-β would provide a more comprehensive understanding of the observed cellular effects. Due to the scope and limitations of the current study, our focus was restricted to evaluating proliferation and apoptosis at the cellular level, considering that we have provided evidence of the mechanism of action in a previous paper (doi:10.3390/ijms19092758). However, we acknowledge the importance of these molecular pathways and intend to address them in future studies.

Comments 2. NPPS-PE is extracted from pigs. Does NPPS-PE have a rejection effect? Can all species be used? Please provide a specific description of its relevant scope of adaptation and adverse reactions

Response 2: We thank the reviewer for this important observation. As with any biologically derived medication, there is a potential risk of immune rejection when NPPS-PE is used, particularly in in vivo applications. Reports in the literature suggest that porcine-derived surfactants, including those currently in clinical use, may occasionally induce mild immune responses; however, the incidence of severe immunogenic reactions is low, typically below 5% in neonatal clinical studies involving porcine surfactants.

It is essential that each medication be evaluated for safety and efficacy in the target species. Our current study was limited to in vitro evaluation in human lung fibroblasts, and we recognize that further preclinical studies in appropriate animal models are necessary to fully assess the potential for species-specific reactions and adverse effects.

Miner points:

Comments 1. The image pixels are too low;

Response 1: We addressed this concern by including the images with improved resolution at 300 DPI and 1100 pixels.

Comments 2.Figure 3 shows only one cell proliferation, and the article also mentions that NHIF has three cell lines. Is this the result of using one cell? Three types of cells, shouldn't they be three small pictures? There are various methods for detecting proliferation, such as CCK-8, EDU, etc; Suggest adding different methods for detection and multi-party verification;

Response 2: We used three different human normal lung fibroblast cell lines, each derived from a different patient. The results shown in Figure 5 represent the average of three independent experiments performed in triplicate across these three cell lines. We added this clarification in the figure legend. Regarding the proliferation assay, we employed the WST-1 method, which is an indirect colorimetric assay equivalent to the CCK-8 method, to assess cell proliferation. While we acknowledge that multiple methods such as EDU could provide complementary data, we focused on WST-1 for this study. We appreciate the suggestion and will consider incorporating additional proliferation detection methods in future studies to further validate our findings.

Comments 3.Figure 4 shows two detection methods, flow cytometry and immunofluorescence. Figure legend only shows flow cytometry.

Response 3: The micrographs shown in Figure 7 are intended as a visual complement to illustrate the findings obtained by flow cytometry. For this purpose, we used fluorescence microscopy with cells stained using the same fluorophores employed in the cytometry experiments. We apologize for not having clearly stated this in the original version. We have revised the figure legend to clarify this point.

Comments 4.In Figure 5, RT-PCR was used to verify the decrease in expression of Col1A1. Why not use protein level validation such as WB?

Response 4: We appreciate the reviewer’s insightful comment. Indeed, validating the decrease in Col1A1 expression at the protein level, such as by Western blot, would provide complementary evidence. However, considering that we have previously  provided evidence that the reduction in mRNA expression follows a corresponding decrease in protein synthesis (doi:10.1371/journal.pone.0134564), and due to limitations in sample availability and resources, we focused on gene expression analysis in this study.

Comments 5.The HE and Masson staining in Figure 6 showed the most significant effect after 3 days of NPPS-PE administration, which differed greatly from the results in Figure 6B.

Response 5: The HE and Masson staining images in Figure 8C are representative and intended to qualitatively illustrate histological changes. In contrast, Figure 8B presents quantitative results obtained from two independent and complementary methods: (1) hydroxyproline assay, which measures collagen content biochemically, and (2) image analysis of all histological sections using Orbit Image Analysis software for objective quantification. The attenuation of fibrosis is only seen when the treatment is given during the inflammatory phase. We revised the figure legend to clarify that the experimental design was intended to evaluate the effect of treatment initiation timing, and we replaced the day 3 micrograph with a more representative image.

Round 2

Reviewer 1 Report

Comments and Suggestions for Authors

The authors have revised their manuscript in accordance with the suggestions.